# Mechanisms Leading to Gut Dysbiosis in COVID-19: Current Evidence and Uncertainties Based on Adverse Outcome Pathways

**DOI:** 10.3390/jcm11185400

**Published:** 2022-09-14

**Authors:** Laure-Alix Clerbaux, Julija Fillipovska, Amalia Muñoz, Mauro Petrillo, Sandra Coecke, Maria-Joao Amorim, Lucia Grenga

**Affiliations:** 1European Commission, Joint Research Centre (JRC), 21027 Ispra, Italy; 2Independent Researcher, 6000 Ohrid, North Macedonia; 3European Commission, Joint Research Centre (JRC), 2440 Geel, Belgium; 4Seidor Italy SRL, 20219 Milano, Italy; 5Instituto Gulbenkian de Ciência, 2780-156 Oerias, Portugal; 6Católica Medical School, Católica Biomedical Research Centre, Universidade Católica Portuguesa, 1649-023 Lisbon, Portugal; 7Département Médicaments et Technologies pour la Santé, Commissariat à l’Énergie Atomique et Aux Énergies Alternatives (CEA), Institut National de Recherche pour l’Agriculture, l’Alimentation et l’Environnement (INRAE), Université Paris-Saclay, 30200 Bagnols-sur-Cèze, France

**Keywords:** SARS-CoV-2 infection, COVID-19, gut dysbiosis, microbiota, gastrointestinal disorders, intestinal inflammation, ACE2 dysregulation

## Abstract

Alteration in gut microbiota has been associated with COVID-19. However, the underlying mechanisms remain poorly understood. Here, we outlined three potential interconnected mechanistic pathways leading to gut dysbiosis as an adverse outcome following SARS-CoV-2 presence in the gastrointestinal tract. Evidence from the literature and current uncertainties are reported for each step of the different pathways. One pathway investigates evidence that intestinal infection by SARS-CoV-2 inducing intestinal inflammation alters the gut microbiota. Another pathway links the binding of viral S protein to angiotensin-converting enzyme 2 (ACE2) to the dysregulation of this receptor, essential in intestinal homeostasis—notably for amino acid metabolism—leading to gut dysbiosis. Additionally, SARS-CoV-2 could induce gut dysbiosis by infecting intestinal bacteria. Assessing current evidence within the Adverse Outcome Pathway framework justifies confidence in the proposed mechanisms to support disease management and permits the identification of inconsistencies and knowledge gaps to orient further research.

## 1. Introduction

Coronavirus disease 2019 (COVID-19) caused by severe acute respiratory syndrome coronavirus 2 (SARS-CoV-2) is still a global public health emergency. A better understanding of the mechanisms underlying the progression and severity of the disease is needed. Particularly, COVID-19 is markedly heterogeneous in terms of clinical outcomes, with a high variation at the individual level. Poor clinical outcomes in COVID-19 patients were notably associated with elderliness and certain pre-existing medical conditions, including but not limited to diabetes, cardiovascular diseases, obesity, and high LDH levels [1,2,3,4,5]. Older age and the comorbidities mentioned above are associated with alterations in the gut microbiota [6,7,8]. Besides, COVID-19 patients exhibit fecal microbiome alterations compared to controls [9,10,11,12]. These changes correlated to COVID-19 severity [12]. Gut dysbiosis, defined as a reduction in gut microbiota diversity or the depletion of beneficial bacteria with an enrichment of the pathogenic ones, may alter susceptibility to SARS-CoV-2 infection [13,14,15]. This is aligned with the evidence that many pathophysiological dimensions of diseases are underpinned by the gut microbiota, especially in chronic inflammatory diseases [16] such as inflammatory bowel diseases (IBD). Although the exact etiologies of IBD remain uncertain, many studies have provided important insights into the central role of gut dysbiosis and barrier dysfunction in inflammatory status [17,18]. The gut microbiota plays an essential role in the education and functions of both the local and systemic immune systems. Besides, emerging evidence has demonstrated important cross-talks between the gut microbiota and many other organs via communication axes such as the gut–lung [19], gut–liver [20,21], and gut–brain [22] axes. Notably, gut dysbiosis during respiratory viral infection has been shown to worsen pulmonary symptoms [23]. Similarly, gut dysbiosis and disrupted intestinal barrier can cause neurological inflammation [22] or hepatic inflammation through the translocation of endotoxins and bacteria via the portal vein [24]. Consistently, taking into account gut microbiome-mediated mechanisms may help depict a comprehensive overview of COVID-19 pathogenesis. Exploring how gut dysbiosis as a pre-existing condition in some COVID-19 patients mechanistically influences the disease progression and impacts the clinical outcomes might help identify high-risk patients, and has been discussed elsewhere [5]. Here, we aim to investigate how SARS-CoV-2 might directly alter the gut microbiota, thus considering gut dysbiosis as a direct consequence of the virus in the gastrointestinal (GI) tract. Recently, animal studies have provided evidence for a direct impact of SARS-CoV-2 infection on the gut microbiota. A study conducted in transgenic mice expressing human ACE2 showed that the gut microbiome is affected by SARS-CoV-2 in a dose-dependent manner after intranasal inoculation [25]. In Syrian hamsters, SARS-CoV-2 infection was associated with mild intestinal inflammation, relative alteration of the intestinal barrier property, and alteration of the gut microbiota [26]. SARS-CoV-2 infection in nonhuman primates was associated with changes in the gut microbiota composition and functional activity [27]. However, despite the dynamic research, the underlying pathways leading to gut dysbiosis in COVID-19 are still poorly understood.

To contribute to deciphering these mechanisms, the Joint Research Centre of the European Commission initiated an interdisciplinary project, the CIAO project, to model the pathogenesis of COVID-19 using the Adverse Outcome Pathway (AOP) framework [28,29,30,31]. The AOP approach is well established in regulatory toxicology [32] but is innovatively applied here to a viral disease of high societal relevance. The project relies on the assumption that an AOP-driven organization of the relevant knowledge will improve the integration of the tsunami of data on COVID-19 [28]. The AOP approach does not capture all the details in a biological pathway, but aims for a pragmatic identification of successively linked key events (KE) that represent essential steps in a pathway leading to an adverse outcome [33,34,35,36]. A key event describes a measurable and essential change in a biological system that can be quantified in experimental or clinical settings [32]. The AOP framework also provides a structured approach for the evaluation of the level of evidence currently available to ascertain the causal relationships between pairs of successive key events [37]. AOPs do not build on the correlation between two events but gather and weigh the evidence for their causal relationship. Because of this mechanistic and causal description of the pathways, AOPs help elucidate the pathophysiological mechanisms also by learning from other diseases, such as IBD or respiratory virus-related diseases presenting gut dysbiosis. Finally, an AOP integrates knowledge across the different biological levels (from molecular, cellular, tissue, organ, and up to organism level). While research tends to compartmentalize in silos, this pandemic calls for an interdisciplinary integration of data from the different experimental systems. Hence, the AOP approach allows the structured review and organization of rapidly growing relevant *in vitro*, *in vivo*, and clinical data. Assessing the evidence currently available using the AOP framework permits the identification of critical inconsistencies and knowledge gaps guiding future research needs. The AOPs are steered by the Organization for Economic Co-operation and Development (OECD), which maintains a centralized online platform called AOP wiki (https://aopwiki.org/ accessed on 29 June 2022), where information captured in AOPs is openly accessible. Numbers in the text refer to these AOP-wiki pages (Table 1).

This study was conducted as part of the CIAO project (https://www.ciao-covid.net/ accessed on 29 June 2022) aiming to provide a holistic overview of the COVID-19 pathogenesis through the Adverse Outcome Pathway framework, offering scientists from different fields an international platform to collaborate across disciplines [1]. Here, we outlined three putative pathways initiated by SARS-CoV-2 presence in the gut leading to gut dysbiosis. We applied the AOP approach to analyze the weight of available evidence supporting the causality of the key event relationships (KER) involved in the proposed pathways. For each causal step, we first described the biological plausibility, then we explored the existing literature and data qualitatively and quantitatively supporting this link, and finally, we highlighted the current inconsistencies, uncertainties, and knowledge gaps. Ultimately, we discussed the potential implication of each pathway for disease management.

## 2. Enteric SARS-CoV-2 Presence Leads to Intestinal Inflammation Altering Gut Microbiota

### 2.1. SARS-CoV-2 Entry into Enterocytes Leads to Intestinal Inflammation

The biological plausibility, evidence, and uncertainties for a productive SARS-CoV-2 enteric infection (an active replication in the GI tract) inducing intestinal inflammation are described in detail elsewhere [38]. Briefly, following binding to the ACE2 receptor (KE1739), SARS-CoV-2 enters into enterocytes (KE1738) and might replicate (KE1847) after antagonizing the antiviral response (KE1901). Viral infection induces the secretion of pro-inflammatory mediators (KE1493), which recruit inflammatory cells (KE1497). SARS-CoV-2 enters into enterocytes via binding to the ACE2 receptor and cleavage, preferentially by transmembrane serine protease 2 (TMPRSS2) at the plasma membrane. Enterocytes in the small intestine express the highest levels of ACE2 in the human body [39,40], and co-express TMPRSS2, indicating potential enteric infection [39,41,42]. ACE2-KO intestinal organoids were fully resistant to SARS-CoV-2 infection [43], suggesting that ACE2 is the entry receptor for SARS-CoV-2 in intestinal cells *in vitro*. Following cellular entry, SARS-CoV-2 induces an antiviral response [44]. The timely production of type I interferons by host cells is critical for limiting viral replication and promoting antiviral immunity [45]. While a body of evidence points towards a productive enteric infection, it is not firmly established that SARS-CoV-2 can actively replicate in the human intestine [38]. Specific conditions, such as viral load, comorbidities, age, medication, inflammatory status, fasted–fed status, or pre-existing dysbiosis, might render the GI epithelium permissive to SARS-CoV2 infection [38]. In addition to interferon, viral infection induces the release of pro-inflammatory cytokines, such as interleukins (IL) and tumor necrosis factor (TNF) alpha by epithelial cells [27,46]. Pro-inflammatory signaling recruits immune cells to the gut. This local inflammatory response due to viral entry into cells and potential active replication might alter the gut microbiota (Figure 1).

### 2.2. Binding to Enteric ACE2 Leads to Intestinal Inflammation

Functional studies based on colitis animal models have indicated that the modulation of ACE2 expression itself affects the severity of intestinal inflammation. ACE2 deficiency causes enhanced susceptibility to dextran sodium sulfate-induced colitis [47], suggesting that ACE2 plays a protective role in colitis. Moreover, Ang (1–7) treatment alleviates colitis progression, whereas the blockade of Mas aggravates the disease [48], indicating the protective role of the ACE2/Ang (1–7)/Mas axis. In contrast, treatment with the ACE2 inhibitor GL1001 reduces the severity of colitis [49], suggesting that ACE2 plays a pathogenic role in intestinal inflammation. During SARS-CoV-2 infection, the downregulation of ACE2 would potentially result in unopposed functions of Ang II and decreased levels of Ang (1–7), thereby shifting the balance toward the pro-inflammatory side [50,51]. In IBD, reduced small bowel but elevated colonic ACE2 levels are associated with inflammation, suggesting compartmentalization of ACE2-related biology in the small intestine and colon inflammation [52]. Further studies are needed to assess if intestinal ACE2 dysregulation due to the interaction with SARS-CoV-2/S proteins leads to intestinal inflammation.

### 2.3. Intestinal Inflammation Leads to Alteration of Gut Microbiota

*Biological plausibility.* Intestinal inflammation is associated with aerobic conditions, biological sources from dying epithelial cells, and mucus thickness, which provide an optimal environment for the growth of microorganisms. Dysbiosis is defined as a reduction in microbial diversity and a combination of the loss of beneficial bacteria and a rise in pathogenic ones (KE1954).

*Evidence*. Plasma concentrations of inflammatory cytokines correlated with gut microbiota composition in studies on COVID-19 patients [10]. Several studies in other diseases provided evidence that an inflamed gut microenvironment induces gut microbiota alterations [53,54,55,56,57]. The alteration is often characterized by blooms of normally low-abundance and harmful bacterial species (e.g., *Enterobacteriaceae)* that are capable of utilizing nutrients found more abundantly in the inflamed gut, while other families of symbiotic bacteria succumb to the inflammatory environmental changes [53,54,55,56,57].

*Feedback loop.* A causal role for gut microbes in generating an inflammatory phenotype was demonstrated. Germ-free mice receiving microbial transfers from insulin-resistant mice exhibited more inflammation than mice receiving microbial transfers from controls [58]. In humans, abundant reports highlight the role of gut microbiota in the pathogenesis of inflammatory diseases such as asthma, type 1 and type 2 diabetes mellitus, and obesity [59,60,61]. Antibiotic-resistant *Klebsiella* species can lead to inflammation in genetically susceptible hosts [62]. Adherent-invasive *E. coli*, commonly reported as enriched in IBD, increases chemokine secretion (IL-8/CCL20 levels). Other *Enterobacteriaceae* species, namely *Citrobacter rodentium* and *Salmonella*, utilize virulence factors to induce intestinal inflammation, which subsequently confers a growth advantage for these pathogens in the intestinal lumen to compete with beneficial bacteria [63,64,65]. Some bacteria produce short-chain fatty acids with anti-inflammatory properties [66,67,68]. *Faecalibacterrium pausnitizii,* reduced in IBD [69], convert acetate to butyrate, which facilitates the regeneration of colonocytes, thus maintaining intestinal integrity [49] and the balance between Th7 and Treg cells to prevent intestinal inflammation [70]. The reduction of butyrate-producing bacteria contributes to intestinal inflammation; notably, T_reg_ cells were shown to be activated by butyrate, blocking an excessive proinflammatory response [71]. Butyrate can exert an anti-inflammatory effect in part by suppressing the activation of NF-κB [72], a transcription factor that regulates the inflammatory and innate immune responses [73]. In addition, butyrate strongly inhibits the interferon-gamma (IFN-γ) signaling to ameliorate inflammation [74]. Butyrate also targets peroxisome proliferator-activated receptor-γ (PPARγ) to prevent colon inflammation [75]. IL-10-deficient mice developing enterocolitis when maintained in conventional conditions showed no evidence of colitis when kept in a germ-free environment, suggesting that resident enteric bacteria are necessary for immune system activation in these mice [47]. Anaerobic and mutually exclusive *Bacteroides* species could dominate the microbiota and exert commensal, mutualistic, or pathogenic behaviors depending on host–microbe interactions, bio-geographical location, and nutritional availability. As a known pathobiont in IBD, *Bacteroides vulgatus* activates NF-kB pathways, and some strains are important for colonization and persistence in CD. Similarly, entero-toxigenic *B. fragilis* has been shown to promote intestinal inflammation and possibly colon carcinogenesis through the activation of NF-kB [76], resulting in increased pro-inflammatory cytokine levels, such as IL-8/CXCL8. The composition of the gut microbiome has been associated with the severity of COVID-19, possibly via its immune-modulatory properties. Gut commensals with known immunomodulatory potential, such as *F. prausnitzii*, *Eubacterium rectale*, and *Bifidobacterium adolescentis*, were found to be significantly under-represented in COVID-19 patients compared with healthy controls, and were associated with disease severity after taking account of antibiotic use and patient age [10]. Furthermore, the microbial imbalance found in COVID-19 patients was also associated with raised levels of inflammatory cytokines such as C-reactive protein The inflammatory phenotype could represent a risk factor in SARS-CoV-2 infection.

*Uncertainties, inconsistencies, and gaps.* Disturbances of the microbiota are evident in inflammatory diseases. However, despite encouraging evidence from animal models in which inflammatory conditions were successfully treated via gut microbiota manipulation, data from human trials are less conclusive. It is still unclear from human studies whether the alteration in the microbial community is a cause or consequence of inflammation. A higher degree of resolution of microbiome analysis matched with lifestyle factors, heterogeneity of the host genotype, and epigenome, is likely to be required to advance the understanding of host–microbe interactions.

The gut microbiota also play an important role in the production of interleukin-22 (IL-22) in the gut, which is central to the induction of antimicrobial peptides, and promotes the protective functions of the epithelial barrier [50]. Klooster et al. [77] showed that intestinal viral infections induce IL-22 expression by T cells stimulated by IFNβ1-mediated IL-7 production by epithelial cells and IL-6 production by fibroblasts. Their findings suggest that IL-22 modulates genes involved in viral entry and replication. Specifically, IL-22 inhibits the expression of the viral entry receptors, ACE2 and TMPRSS2, while increasing the expression of antiviral proteins [77]. Although IL-22 is well-known for its role in bacterial defense, there are limited and conflicting data on the importance and regulation of IL-22 in intestinal viral defense.

### 2.4. Potential Implication for Disease Management

The persistence of gut microbiota dysbiosis after disease resolution in COVID-19 could contribute to persistent symptoms, highlighting a need to understand how gut microorganisms are involved in inflammation and COVID-19. Notably, it remains unknown whether inflammation-associated gut microorganisms enriched in COVID-19 play an active part in the disease or flourish opportunistically due to a depletion of other gut microorganisms. Follow-ups of patients with COVID-19 (e.g., 3 months to 1 year after clearing the virus) are needed to address questions related to (i) the duration of gut microbiota dysbiosis post-recovery, (ii) the link between microbiota dysbiosis and long-term persistent symptoms, and (iii) whether the enrichment/depletion of specific gut microorganisms predisposes recovered individuals to future health problems.

## 3. Intestinal ACE2 Dysregulation Inducing Gut Dysbiosis

Research in the last few years has highlighted the key role of ACE2 in intestinal homeostasis by influencing multiple processes [78,79,80,81], including the modulation of gut microbiota [82,83,84]. Therefore, it is plausible that the binding of the viral S protein to ACE2 in the gut may lead to the dysregulation of physiological functions such as alteration of the GI amino acid (AA) metabolism, altering the gut microbiota (Figure 2).

### 3.1. Binding of S Proteins to Enteric ACE2 Induces Intestinal ACE2 Dysregulation

*Biological plausibility*. The binding of S proteins to ACE2 is likely to impede the physiological functions of ACE2. In the gut, ACE2 modulates the local renin–angiotensin signaling (RAS) system in a paracrine and autocrine manner, mediating cell-specific growth, proliferation, and metabolic activity [83]. Little is known about the potential role of intestinal ACE2 in modulating the local KKS system in the gut. The role of ACE2 as a chaperone for neutral AA transporters in the intestines is its most studied RAS-independent and non-enzymatic function [85,86]. ACE2 and ACE, components of the RAS system, are present in the intestine. Intestinal ACE2 stabilizes the transporter B^0^AT1 (Slc6a19), which mediates the uptake of neutral dietary AA, such as tryptophan (Trp), into intestinal cells in a sodium-dependent manner [86]. ACE2 has also been proposed to interact functionally with sodium-dependent imino transporter 1 (SIT1), a luminal L-proline transporter expressed in small intestine enterocytes [78]. Therefore, it is plausible that SARS-CoV-2 binding might interfere with ACE2 association with AA transporters and their function.

*Evidence.* Extensive evidence exists for ACE2 dysregulation as a result of interaction with the viral S proteins in different cells and tissues, with changes in ACE2 mRNA expression, protein levels, and enzymatic activity (KER2311). Evidence for ACE2 protein down-regulation mediated by viral S proteins comes from lung and liver-derived cell systems [79,80] which monitored ACE2 protein levels in whole cell lysates. Membrane and cellular ACE2 protein down-regulation following treatment with SARS-CoV-1 S protein have also been demonstrated in studies with kidney cell lines that concomitantly monitored and showed increased ACE2 enzymatic activity in the extracellular compartment [81,87,88]. In these non-GI test systems, the decrease of the full-length ACE2 cellular protein is due to S-protein-mediated cleavage of ACE2 by cellular proteases (TACE/ADAM17). The precise role of ACE2 cleavage and shedding in SARS-CoV and SARS-CoV-2 viral entry and/or maturation of infective particles remains to be elucidated. ACE2 down-regulation at the transcriptional level has been reported in kidney biopsies from deceased patients [89] and in GI tract-derived organoids [90]. Using a single-cell transcriptomics approach and multiplex single-molecule RNA fluorescence in situ hybridization (FISH), ACE2 mRNA down-regulation was observed in both ileal- and colon-derived 2D organoids infected with SARS-CoV-2 (relative to mock infected organoids) [90]. Tissue-specific differences were noted. In ileum-derived organoids, ACE2 mRNA was down-regulated in the bystander cells (cells not showing active SARS-CoV-2 replication as judged by detection of viral RNA) whereas in the colon-derived bystander cells, ACE2 was comparable to mock/uninfected cells. This difference may also be due to the method applied to determine the threshold for distinguishing actively infected and bystander cells (which also contained detectable viral RNA). Nataf and Pays (2021) [91] reported profound but transient down-regulation of ACE2 mRNA in SARS-CoV-2-infected differentiated human intestinal organoids compared to controls. Interestingly, they also reported decreased B^0^AT1 mRNA, which requires ACE2 for its membrane expression and function [86]. The mRNA levels for both ACE2 and B^0^AT1 returned to baseline by 60 hpi. Nataf and Pays [91] re-analyzed the mRNA expression levels generated in a study by Lamers et al. [46] but reported results from earlier time points.

Evidence for up-regulation of ACE2 (mRNA and protein) following interaction with SARS-CoV-2 S proteins is available in a significant number of studies with non-GI-infected tissues or *in vitro* cell systems (KER2311). In differentiated human small intestinal 3D organoids (DIF) infected with SARS-CoV-2, a modest ACE2 mRNA up-regulation was reported [46]. The DIF showed significantly higher levels of ACE2 expression compared to expanding organoids (EO). Data in this study shows ACE2 mRNA down-regulation in the DIF organoids [91]. SARS-CoV infection up-regulated ACE2 mRNA at 24 and 60 hpi in EO while the data for SARS-CoV-2 in EO showed up-regulation of ACE2 mRNA at 60 hpi only. Up-regulation of both ACE2 mRNA (~3×) and protein (1.3×) in 2D differentiated Caco-2-derived infected with SARS-CoV-2 were observed compared to uninfected cells [78]. Up-regulation was noted when the viral titer was at saturation. ACE2 mRNA up-regulation was also reported with SARS-CoV-2 in human colon 3D organoids [92].

*Uncertainties, inconsistencies, and gaps.* Evidence supports the high plausibility of ACE2 dysregulation in the GI tract due to the interaction with SARS-CoV-2. However, direct evidence for ACE2 dysregulation resulting from S protein binding rather than viral replication in the gut or in gut-derived systems is currently lacking. In addition, there are inconsistencies in the evidence that need further consideration.

The apparent inconsistencies regarding the direction and magnitude of ACE2 dysregulation in the different studies (using various test systems) may reflect the dynamic and temporal components of the dysregulation. The latter could be driven not only by the interaction of ACE2 with the surface viral components, but also by the interaction of the replicating viral components with the innate immunity response elements, particularly in the test systems using replicating viruses. ACE2 mRNA down-regulation in SARS-CoV2-treated GI-derived organoids was reported in enterocytes actively replicating the virus [90]. A second study also reported profound but transient ACE mRNA downregulation [91]. Contrary evidence for SARS-CoV2 mediating up-regulation of ACE2 mRNA in GI organoids [46,92,93] is consistent with similar studies in many other tissue/organ systems (KER2311) and with the finding that *ace2* is an Interferon Stimulated Gene (ISG) in airway epithelial cells [94] and in colon enterocytes [92]. These studies also demonstrated a time concordance of ACE2 mRNA up-regulation with stimulation of ISG response in the infected organoids [46,92,93]. Interestingly, a scRNAseq study by Triana et al. [90] found that SARS-CoV2 exposure induced distinct proinflammatory and ISG expression profiles in infected and bystander cells in the organoid. Expression of ISGs was pronounced in bystander cells, while the infected cells showed strong NFkB/TNF-mediated pro-inflammatory response but limited production of ISGs. This suggests that while SARS-CoV-2 may activate ISG by paracrine signaling, it may suppress the autocrine action of interferon *i.e* induction of ISG, including ACE2, in infected cells. This would be consistent with ACE2 down-regulation in the infected cells observed in this study. In addition, this may explain why in some studies ACE2 mRNA down-regulation can be observed under certain conditions and at some (earlier) time points of replication. Furthermore, the causal relationship between an observed increase in ACE2 mRNA and dysregulation at protein and enzymatic levels remains to be elucidated. Indeed, most recently Harnik et al. [95] examined the spatial discordances between mRNAs and proteins in the intestinal epithelium and their significance for the interpretation of transcriptomic data. In addition, in the intestines of SARS-CoV-2-infected Hamsters, mRNA expression of ACE2 was up-regulated and ACE2 function was decreased [26]. Such apparent discordances have also been reported in the heart and lung tissue of mice and humans (KE1854).

The identification of alternative forms of ACE2 mRNA and protein, an N-terminus truncated dACE2, which appears to have a distinct transcriptional regulation profile compared to flACE2 [95,96,97], may also account for some of the observed inconsistencies. A detailed analysis of experimental conditions in the past, and careful design of probes and primers in future studies would be informative. Interestingly, concomitant down- and up-regulation of 97kD and 80kD anti-ACE2 polyclonal Ab-reacting proteins have been detected in human colon adenocarcinoma cell line HT29 [98]. Considering only one form of ACE2 relevant, the authors concluded that ACE2 was down-regulated in mature differentiated enterocytes compared to undifferentiated ones. This is in contrast to all the studies described above which demonstrated that the highest level of ACE2 (both mRNA and protein) were detected in the mature enterocytes and at the brush borders of the intestine and 3D organoids [46,90,92,93]. The inconsistencies disused above clearly illustrate the need for careful characterization of the test systems to facilitate robust interpretation of the results.

The majority of studies have focused on ACE2 mRNA levels, while protein and functional analyses are often lacking, particularly in the GI system. The novel gut-derived organoid systems could help address this gap by monitoring the level and cell distribution of ACE2 protein as well as its function as B^0^AT1 chaperon, by monitoring the membrane expression and the transporter function of B^0^AT1. In addition, treatment with S protein and with non-replicating SARS-CoV-2 pseudoviruses [98] may better address any potential direct effect of S-binding on ACE2 dysregulation. Indeed, it remains to be elucidated if S protein alone can elicit ACE2 dysregulation, as this would mean that a nonviable virus reaching the gut lumen would be sufficient to induce such a mechanism. Finally, a development of more complex organoid systems that would also include microbiota and/or elements of the immune system is needed to better examine ACE2 dysregulation by SARS-CoV2, but also the effects of such dysregulation at higher organizational levels and in conjunction with the other elements of the RAS system. Finally, evidence of up- or down-regulation of ACE2 in the GI tract of SARS-CoV-2-infected patients was not available. Examining potentially existing or generating GI-specific transcriptomic, proteomic and biomarker databases of COVID-19 patents may help address some of these uncertainties. This again highlights the importance of interdisciplinary collaboration between basic, translational, and clinical researchers.

### 3.2. Enteric ACE2 Dysregulation Leads to Gut Microbiota Alteration

*Biological plausibility.* ACE2 co-expresses with several AA transporters in enterocytes, such as B^0^AT1 for Trp [99] and SIT-1 for proline [78,100,101]. Thus, in the gut, ACE2 modulates dietary AA transport. Trp regulates the secretion of antimicrobial peptides by Paneth cells through the mTOR pathway [102]. Those antimicrobial peptides impact the composition and diversity of the microbiota [12,103]. In addition, the gut microbiota is influenced by the host intestinal AA metabolism as bacteria of the gut use dietary AA for protein synthesis [104,105,106]. Alteration of dietary AA transport due to viral S proteins binding to ACE2 could modify the ratio of AA-fermenting bacterial species and their metabolic pathways. Metabolism of AA by gut bacteria results in the formation of diverse metabolites, several of which are considered deleterious (nitrosamines, heterocyclic amines, and hydrogen sulfides), while others are beneficial, such as short-chain fatty acids (SCFA), namely butyrate, propionic acid, and acetic acid. Finally, as a regulator of local RAS, ACE2 receptor hijacked by the viral S proteins could lead to reduced ACE2 cleavage of AngII, an increase in local Ang II levels, and Ang 1-7 decrease resulting in luminal activation of ATR1 [107], enhancing permeability [99], and impacting gut microbiota. In addition, the GI RAS appears to be involved in numerous processes in the gut including AA, fluid, and electrolyte absorption and secretion [108].

*Evidence.* Some evidence linking ACE2-mediated altered dietary AA (such as Trp) and gut dysbiosis exist. Ace2 KO mice lack B^0^AT1 [86] and exhibited reduced Trp serum levels, along with downregulated expression of the mTOR pathway, inducing impaired expression of small intestinal antimicrobial peptides, and resulting in altered gut microbiota, which was re-established by Trp supplementation [109]. Exacerbated diabetes-induced dysbiosis was also observed in ACE2 KO/y-Akita mice [110]. ACE2 is also a co-receptor of SIT-1 transporting proline. ACE2 KO mice showed decreased intestinal proline absorption [111,112], not reflecting an increase of intestinal permeability but an alteration of the selective aspect of the intestinal barrier. In fecal microbiota of COVID-19 patients, the abundance of opportunistic pathogens was higher, and SCFA-producing bacterial populations were lower compared to healthy controls [9,113], suggesting that intestinal AA metabolism is altered. There is preclinical evidence of the presence of all RAS components in the GI tract [108,114]. Evidence also indicates a complex association between gut microbiota, ACE2 expression, and Vitamin D in COVID-19 severity. Vitamin D contributes to the regulation of the gut microbiome by maintaining microbial diversity and by promoting the growth of beneficial commensal strains of *Bifida* and *Fermicutus*. In addition, Vitamin D is a negative regulator for renin expression and interacts with the RAS/ACE/ACE-2 signaling axis [115].

*Feedback loop.* Interestingly, the gut microbiota seems to influence Ace2 expression and activity. A study found that several *Streptococcus* spp. increased the level of ACE2 protein in mammalian cells [116]. In patients, *Coprobacillus* enrichment—associated with clinical severity of COVID-19 [12] has been shown to upregulate colonic ACE2 in mice [48,117]. The abundance of specific gut bacteria such as certain Bacteroides species (*Bacteroides dorei*, *Bacteroides thetaiotaomicron*, *Bacteroides massiliensis*, and *Bacteroides ovatus*) was associated with a reduction in ACE2 expression in the mouse gut [48,117] and negatively correlated with fecal SARS-CoV-2 load [12], suggesting that they may limit the ability of SARS-CoV-2 to enter enterocytes [48]. In addition, gnotobiotic rats colonized with 9 bacterial phyla showed a decrease in colonic Ace2 expression compared to germ-free rats [118].

*Uncertainties, inconsistencies, and gaps.* Evidence linking altered levels/functions of ACE2 with altered uptake of dietary AA (Trp and/or proline) and alteration in gut microbiota needs to be further evaluated. One could examine Trp levels in ACE2-infected mice and assess dysbiosis with or without Trp supplementation. The use of S proteins, non-replicating SARS-CoV-2 pseudoviruses, or SARS-CoV-2 viruses might be informative as well as further exploration of the pro- and prebiotic effects on ACE2 regulation. In addition, not enough evidence is available so far regarding the intestinal RAS following ACE2 dysregulation in COVID-19.

### 3.3. Potential Implications for Disease Management

Many forms of diarrheal disease depend on the dysregulation of intestinal ion transporters, and an imbalance between secretory and absorptive functions of the intestinal epithelium [119]. It is tempting to consider that infectious dysbiosis and diarrhea might be effectively targeted by small molecules that act specifically on transporters implicated in the disease. Indeed, efforts are already ongoing to identify such molecules and some promising candidates have been identified [120,121], but they seem to be focused on exploring the ACE2–Spike protein interaction rather than ACE2 function/activity alone, or RAS-related function which is critical for cardiovascular homeostasis [122]. Similar approaches screening for the intestinal-specific functions of ACE2 may help in the management of gut dysbiosis during COVID-19 and potentially other ACE2-mediated gut dysfunctions. Based on the AOP outlined above, screening for modulating factors of ACE2 function that alter gut microbiota would be an informative target focus. Recognizing that testing or modeling systems that include microbiota are not yet fully available, the evidence analysis in the AOP justifies efforts needed for their development.

## 4. SARS-CoV-2 Infection of Microbial Bacteria Driving Gut Dysbiosis

In the gut, human cells might not be the sole SARS-CoV-2 targets. SARS-CoV-2 infection of human gut bacteria might be another mechanism driving dysbiosis in COVID-19 patients (Figure 3).

### 4.1. Viral Entry in Gut Bacteria Leads to Coronavirus Production in Bacteria

*Biological plausibility.* Bacteriophages are viruses that infect prokaryotic hosts, such as bacteria of the gut microbiome. Typically, a phage virion binds to the host cell surface using a phage receptor-binding protein triggering the insertion of its genome into the host [123].

*Evidence*. A series of serine protease TMPRSS2 and peptidyl peptidase with high similarity to ACE2 peptidase domain were identified in silico in bacteria of the *Proteobacteria* phylum [124]. Transmission electron microscopy analysis showed the presence of SARS-CoV-2 particles on the surface and inside gut bacteria obtained from COVID-19 patients [125], consistent with a viral tropism for gut bacteria [13]. SARS-CoV-2 replication has been observed outside the human body in bacterial growth medium, following bacterial growth, and reduced by antibiotics administration [13].

*Uncertainties, inconsistencies, and gaps.* A virus able to infect at the same time eukaryotic and prokaryotic cells has never been described before. In addition, enveloped bacteriophages are not very common. The best-known family (*Cystoviridae*) is lipid-containing with three double-stranded RNA (ds-RNA) genome segments: resembling the family *Reoviridae*, cystoviruses served as a simple model for reovirus assembly, but *Cystoviridae* genome packaging mechanism have not yet fully elucidated [126]. Looking for taxa potentially acting as a receptor for the virus would be really informative. Evidence of two bacterial species susceptible to being infected by SARS-CoV-2 has been recently reported [125]. However, the full picture of SARS-CoV-2-susceptible human gut bacterial species is lacking [13].

### 4.2. Coronavirus Production in Gut Bacteria Leads to Alteration of Gut Microbiota

*Biological plausibility.* Bacteriophages can shape bacterial communities by predation or by horizontal gene transfer through transduction. Besides, viruses can modulate microbiota function by modulating their metabolism.

*Evidence*. Altered microbiota compositions were found to be independent of the presence of SARS-CoV-2 in the respiratory tract, disease severity, and GI symptoms, but correlated with GI levels of SARS-CoV-2 RNA [127].

*Uncertainties, inconsistencies and gaps.* There is a wealth of literature on the role of bacteriophages within the human gut. However, there are still large areas that require further investigation, and if fully elucidated, could trigger beneficial treatments for human diseases, similar to what we are currently seeing with the bacterial component of the human microbiome [128]. A key issue is that current analysis tends to focus on the known annotated component of viral datasets [129,130] and the need for reproducible methods limiting bias at the different steps. Furthermore, besides taxonomy, investigating the metabolomic alteration following SARS-CoV-2 infection *in vitro* would help to provide evidence of the causal link between bacterial coronavirus production and dysbiosis.

### 4.3. Potential Implications for Disease Management

This proposed mechanism might have a direct effect on human health. If the virus is hosted by bacteria in the gut microbiome, eliminating the bacterial host with appropriate antibiotics might kill the virus [131]. The efficacy of some antibiotics (like rifaximin and azithromycin) in reducing viral RNA load to negligible levels in *in vitro* fecal microbiota cultures obtained from stool samples of SARS-CoV-2-positive individuals has been reported [13,132]. However, a better understanding via the AOP concept of these complex interactions makes prevention or adequate therapeutic interventions mechanism-based, taking into account different modulating factors [123]. In this regard, multidisciplinary approaches that couple tests on the efficacy of antimicrobials with proteomic and electron microscopy image analyses would be beneficial to shed light on the potential viral tropism of SARS-CoV-2 for gut bacteria.

## 5. Central Role of Gut Microbiota in COVID-19 and Potential Modulation

The three above proposed pathways leading to an alteration of gut microbiota following SARS-CoV-2 presence in the gut lumen are non-mutually exclusive but rather interconnected (Figure 4).

### 5.1. Gut Microbiota and Intestinal Barrier Integrity in COVID-19

Together with the mucosal barrier and the cellular immune system, the intestinal epithelial cell monolayer and the tight junction proteins act simultaneously as a physical barrier against harmful external substances, as well as a selective barrier. Increased intestinal permeability, a sign of an impaired barrier function, enhances the translocation of gut bacteria and bacterial toxins from the intestinal lumen into the systemic circulation. The gut microbiota ensures intestinal barrier integrity through diverse mechanisms [48] (Figure 4, dashed grey lines). Beneficial butyrate-producing bacteria are proposed to maintain intestinal integrity, as butyrate, a short-chain fatty acid (SCFA), facilitates the regeneration of healthy colonocytes [49]. A reduced relative proportion of bacteria producing SCFA was observed in Syrian hamsters infected with SARS-CoV-2, compared to non-infected controls, with a transient decrease in systemic SCFA amounts [26]. Decreases in the abundance of butyrate-producing bacteria and a decline in SCFA were observed in severe COVID-19 [10,12,133]. Besides the reduction of beneficial bacteria, the overgrowth of pathobionts, such as *Escherichia coli* or *Salmonella enterica,* disrupts intestinal barrier function [134,135,136]. Outgrowth of pathogenic *Prevotella* has been associated with reduced mucus secretion, one crucial protective layer of the intestinal barrier [137]. Blooms of pathogenic bacteria have been observed in hospitalized COVID-19 patients, along with the translocation of gut bacteria into the blood [25]. Lowered levels of butyrate-producers and higher levels of opportunistic pathogens (including *E. coli* and *S. enterica)* were observed in COVID-19 patients compared with H1N1 patients and healthy controls [9]. In addition, gut microbiota composition correlated with plasma levels of tissue damage markers, altered tight junctions, and microbial translocation in COVID-19 patients [10]. Finally, the colonic mucus barrier is shaped by the composition of the gut microbiota [138]. Alteration of the gut microbiota might contribute to disrupting the mucus barrier.

Human intestinal organoid co-cultures with microbes could represent useful systems to investigate the protective function of bacteria on gut permeability upon SARS-CoV-2 infection [139]. In addition, similar to the treatment of other diseases, treating SARS-CoV-2 infected mice or Syrian hamsters with SCFA supplementation [26,51], prebiotics, or probiotics (such as *Lactobacillus reuteri* in rodents), [140] and evaluating the intestinal permeability (dextran and bacterial translocation) in parallel with microbiota omics could strengthen our understanding of the relationship between gut microbiota and the intestinal barrier in COVID-19 pathophysiology.

### 5.2. Central Role of the Gut (Microbiota) in COVID-19 and Long COVID

Dysbiosis, intestinal inflammation, and leaky gut are intimately interconnected (Figure 4) and intestinal homeostasis is increasingly recognized as an underpinning clinical driver in several noncommunicable diseases as well as in COVID-19. Accumulating evidence supports that altered gut microbiota and associated leaky gut may contribute to the GI symptoms and the cytokine storm and multiorgan complications in COVID-19 [141,142]. In critically ill patients with sepsis and respiratory distress, bacterial translocation is widely documented [143,144]. Higher plasma levels of gut permeability markers were found in COVID-19 patients, along with abnormal presence of gut bacteria in the blood [145,146]. These markers correlated with higher levels of C-reactive peptide (a marker of hyperinflammation) and with a higher mortality rate [146]. Serum levels of lipopolysaccharide-binding protein were higher in patients with severe COVID-19 and were associated with circulating inflammation biomarkers [147]. Altered intestinal homeostasis induces diarrhea [148], which is the digestive symptom most commonly reported in COVID-19 patients [149,150,151,152,153].

Despite the well-documented prevalence of GI symptoms and the high rate of SARS-CoV-2 fecal RNA shedding, the isolation of replication-competent virus from fecal samples has not been reproducibly and systematically demonstrated [38]. The biological, clinical, and epidemiological relevance of SARS-CoV-2 shedding remains unclear [154]. SARS-CoV-2 shedding in stools has been reported from one week to seven months after diagnosis [154,155]. The prolonged presence of viral RNA in feces [154], but not in respiratory samples, and its association with GI symptoms suggests that SARS-CoV-2 infects the GI tract, and that this infection can be prolonged in a subset of individuals with COVID-19. SARS-CoV-2 infection leading to perturbation of the gut microbiome may contribute to the underlying etiology of GI symptoms observed in COVID-19 and long COVID [45,156]. Alteration in the gut microbiome persists long after a patient recovers, suggesting that the gut microbiome may play an important role in long COVID [157]. Long COVID or post-acute COVID-19 syndrome (PACS) is rapidly emerging across the globe and many studies following patients who have recovered from the respiratory effects of COVID-19 identified persistent GI sequelae, including dysbiosis [154,155,158]. While the pathogenesis of long COVID is still under intense investigation, on the four current leading hypotheses [45], it is interesting to note that gut dysbiosis is considered as one of them [157,159]. A comprehensive understanding of the dynamics of fecal clearance of SARS-CoV-2 RNA and its link with gut dysbiosis is currently lacking. Further studies are needed as the gut microbiota could serve as a potential prognosis indicator and could be therapeutically valuable.

### 5.3. Potential Modulation of Gut Microbiota to Mitigate COVID-19

In light of the current insight into the central role of the gut in COVID-19 and long COVID, modulating the gut microbiota to improve disease prevention and management may be relevant. First, fecal microbiota transplantation (FMT) enables stool infusion from a healthy individual to a severely ill patient to restore intestinal microbial balance [160]. So far, FMT has been remarkably successful in the treatment of *Clostridium difficile* infection, but much less in treating other conditions, such as IBD or metabolic disorders. COVID-19 being an infectious disease and not an inflammatory disorder, FMT could be more successful [141]. However, COVID-19 could potentially be transmitted via FMT, particularly from asymptomatic donors who tested negative for the presence of the virus in their respiratory tract but positive in their stools [161]. No cases of COVID-19 transmission through FMT have been reported so far, but only FMT products generated from stools donated before December/November 2019 were used according to the FDA and Hong Kong recommendations, respectively. Secondly, gut microbiota modulation with probiotics, prebiotics, or diet and therapies preventing gut barrier defects may represent easy-to-implement strategies to mitigate COVID-19 [162]. Clinical trials of probiotics with expected anti-inflammatory effects for preventing or treating SARS-CoV-2 infection are currently ongoing [163]. Next-generation probiotics focusing on butyrate-producing bacteria, or simply increasing the daily intake of dietary fiber are proposed as potential beneficial approaches for COVID-19 patients [141]. A few reports cite indirect evidence for the association between probiotics and COVID-19, primarily based on previous coronaviruses and other viral infections [164,165]. The health benefits of prebiotics to the GI tract, including the inhibition of pathogens and stimulation of the immune system, are due to their ability to modulate the composition and activity of human microbiota [166,167,168]. However, to date, there is no information directly linking prebiotics to COVID-19 infections, although an indirect effect may be hypothesized [169]. Thus, using conventional probiotics is not currently warranted, but is considered promising, and a better understanding of SARS-CoV-2 pathogenesis and its mutual effect on gut microbiota is needed. More generally, diet is obviously a factor impacting gut microbiota [170,171,172,173]. Dietary adaptation may be the easiest method to be implemented in the preventive arsenal against COVID-19 and for general health improvement [141].

## 6. Conclusions and Future Perspectives

Here we explored the evidence currently available in the literature supporting that SARS-CoV-2 induces intestinal inflammation, dysregulates intestinal ACE2 physiological functions, and/or infects gut bacteria, as three potential interconnected mechanisms leading to gut dysbiosis in COVID-19. Based on the current insights into the underlying mechanisms, we discussed the potential implications for disease management in infected patients. In addition, the alterations in the gut microbial community are observed long after the respiratory syndrome is resolved, and thus a better understanding of the underlying mechanisms is needed to capture the potentially important role of microbiota in long COVID. The approach applied also permits identifying knowledge gaps and proposes methods to perform further research. Notably, examining potentially existing or generating GI-specific transcriptomic, proteomic, and biomarker databases of COVID-19 patents may help address some of these uncertainties. Large-scale population-based studies are warranted to validate with more confidence these pathways, and intervention studies could help to explore the roles of gut microbiota alteration in COVID-19 pathogenesis. In addition, it remains unclear to what extent the gut microbiota composition as an outcome of COVID-19 is influenced by clinical management due to the variability across COVID-19 treatments. Due to all these current uncertainties, there is a need to continue the interdisciplinary collaboration between basic, translational, and clinical researchers.

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
