# Peer review of "Mechanisms Leading to Gut Dysbiosis in COVID-19: Current Evidence and Uncertainties Based on Adverse Outcome Pathways"

_jcm, 2022, doi:10.3390/jcm11185400_

Round 1
Reviewer 1 Report
The manuscript “Mechanisms leading to gut dysbiosis in COVID-19: current evidence and uncertainties based on putative adverse outcome 3 pathways.” presents a great overview of the potential pathways of COVID-19 infection on gut dysbiosis. Generally, the manuscript is well written with rigorous work and balanced information. The review is interesting and seems sound, but still, there is room for improvement and some changes are required. I hope that the recommended literature (provided for some sections) can fulfil this gap effectively.
The majority of the sentences are large and can be split into smaller sentences to make the language of the manuscript easy. (i.e. line 178—181, lines 304-309, etc). Could make it shorter and easy for non-natives as well.
- line 34-35 – needs references: 10.3390/coatings12010102; 10.1080/19490976.2022.2031840; 10.3390/microorganisms9061318
- line 40 – Besides “diabetes, hypertension, obesity and high LDH levels [1], [2], [3].” these pre-existing medical conditions there are other ones also, please indicate, as indicated in a similar articles: 10.3389/fcimb.2020.575559, 10.1038/s41569-020-0413-9, 10.1038/ s41575-020-0322-8
- line 46 – 48 – maybe the authors could describe better the gut microbiota’s importance with regard to its effect on health as indicated in other subsections like the “gut-liver” axis, but what about the “gut-brain” axis? Please revise: 10.3389/fmed.2022.813204
- line 94 – subsections should be numbered as it can help in the better understanding of the articles, especially as there are many subsections
- line 96 – please indicate where the “SARS-CoV-2 95 enteric infection inducing intestinal inflammation are described”
- The authors should revise the section from line 210 “Emerging evidence demonstrated the cross-talk between the gut microbiota and the lungs, called the “gut-liver” axis” – I think they wanted to say microbiota and the liver, as indicated in a recent relevant study 10.3390/microorganisms10051045.
- some general information regarding pro-, and perbiotics could be beneficial i.e.: 10.3390/molecules26196076; 10.3390/ijerph19031208
- line 227 – which are those recent researches? Please indicate.
The conclusion is well-drawn, with the major findings and how they are addressing the left-behind research gaps and covering current challenges. The Conclusion and Future perspectives sections have been well prepared and concise.
Author Response
Reviewer 1
Comments and Suggestions for Authors
The manuscript “Mechanisms leading to gut dysbiosis in COVID-19: current evidence and uncertainties based on putative adverse outcome 3 pathways.” presents a great overview of the potential pathways of COVID-19 infection on gut dysbiosis. Generally, the manuscript is well written with rigorous work and balanced information. The review is interesting and seems sound, but still, there is room for improvement and some changes are required. I hope that the recommended literature (provided for some sections) can fulfill this gap effectively.
We want to thank Reviewer 1 for his/her constructive comments and criticisms. We carefully addressed in details the different points raised and added references to relevant literature.
Point 1. The majority of the sentences are large and can be split into smaller sentences to make the language of the manuscript easy. (i.e. line 178—181, lines 304-309, etc). Could make it shorter and easy for non-natives as well.
Response: We have carefully reviewed the manuscript to improve its readability, notably by splitting many sentences into smaller sentences and by shortening sentences. The changes are highlighted in the tracked version of the manuscript.
Point 2. line 34-35 – needs references: 10.3390/coatings12010102; 10.1080/19490976.2022.2031840; 10.3390/microorganisms9061318
Response: We added the reference of the article discussing the gut bacterial dysbiosis and instability is associated with the onset of complications and mortality in COVID-19 line 44 of the revised version:
“Gut dysbiosis, defined as a reduction in gut microbiota diversity or the depletion of beneficial bacteria with an enrichment of the pathogenic ones, may alter susceptibility to SARS-CoV-2 infection [13]; (14); 10.1080/19490976.2022.2031840)”.
We did not add the reference 10.3390/coatings12010102 as the article talks about Nanocarriers for Sustainable Active Packaging, nor the reference 10.1080/19490976.2022.2031840 as the article evaluates the infection of SARS-CoV-2 in testes of hamsters.
Point 3. line 40 – Besides “diabetes, hypertension, obesity and high LDH levels [1], [2], [3].” these pre-existing medical conditions there are other ones also, please indicate, as indicated in a similar articles: 10.3389/fcimb.2020.575559, 10.1038/s41569-020-0413-9, 10.1038/ s41575-020-0322-8
Response: The sentence was changed to avoid limiting the list of comorbidities that could be associated to a poor clinical outcome line 38 of the revised version.
“Poor clinical outcomes in COVID-19 patients were notably associated with elderliness and certain pre-existing medical conditions, including but not limited to diabetes, cardiovascular diseases, obesity and high LDH levels [1], [2], [3], 10.1038/s41569-020-0413-9; 10.3390/jcm11154464.” Older age and the comorbidities mentioned above are associated with alterations of the gut microbiota [6], [7], 10.3389/fcimb.2020.575559).
Point 4. line 46 – 48 – maybe the authors could describe better the gut microbiota’s importance with regard to its effect on health as indicated in other subsections like the “gut-liver” axis, but what about the “gut-brain” axis? Please revise: 10.3389/fmed.2022.813204
Response: We agree with the reviewer that those axes are important as well with regard to gut microbiota impact on health. We added a paragraph to mention the gut-brain, gut-lung and gut-liver axes (p.2) along with 5 additional references including the reference proposed at point 7.
“Besides, emerging evidence demonstrated important cross-talks between the gut microbiota and many other organs via communication axes such as the gut-lung (10.1038/s41385-019-0160-6), gut-liver [20] 10.3390/microorganisms10051045) and gut-brain (10.3389/fmed.2022.813204) axes. Notably, gut dysbiosis during respiratory viral infection has been shown to worsen pulmonary symptoms (10.1016/j.jiph.2020.07.003). Similarly, gut dysbiosis and disrupted intestinal barrier can cause neurological inflammation (10.3389/fmed.2022.813204) or hepatic inflammation through translocation of endotoxins and bacteria via the portal vein (10.1002/rmv.2211).”
Point 5. line 94 – subsections should be numbered as it can help in the better understanding of the articles, especially as there are many subsections
Response: We added subsection numbers to increase the readability of the manuscript. Thank you for this comment.
Point 6. line 96 – please indicate where the “SARS-CoV-2 95 enteric infection inducing intestinal inflammation are described”
Response: We investigated the evidence currently available in the literature regarding a productive infection of the gut by SARS-CoV-2 in another paper submitted in parallel to this Special Issue (jcm1820160). We will ask the editor how we have to refer to this parallel and complementary study.
Point 7. The authors should revise the section from line 210 “Emerging evidence demonstrated the cross-talk between the gut microbiota and the lungs, called the “gut-liver” axis” – I think they wanted to say microbiota and the liver, as indicated in a recent relevant study 10.3390/microorganisms10051045.
Response: We apologize for the mistake, we modified the text accordingly. In addition, the sentence was moved to the Introduction section and other gut-organ axes were discussed with associated references as suggested earlier (point 4).
Point 8. some general information regarding pro-, and perbiotics could be beneficial i.e.: 10.3390/molecules26196076; 10.3390/ijerph19031208
Response: Pre and probiotics and they potential application in the context of COVID-19 are discussed in the text in section 5.3. We added the mentioned reference (https://pubmed.ncbi.nlm.nih.gov/34641619/) (p.13) as well as the reference for currently ongoing clinical trials in COVID-19 of probiotics (10.1016/j.nmni.2021.10083) to provide the readers further relevant references.
Point 9. line 227 – which are those recent researches? Please indicate.
Response: This was an introductory section, the research is cited and described under the Biological plausibility section. We revised the introductory section by including references (79) up (82) as well as new references line 240 of the revised version.
« Research in the last few years highlighted the key role of ACE2 in intestinal homeostasis by influencing multiple processes [79]-[82], including the modulation of gut microbiota (10.1053/j.gastro.2020.07.067; 10.1152/ajpgi.00099.2021; 10.1038/nature11228). »
The conclusion is well-drawn, with the major findings and how they are addressing the left-behind research gaps and covering current challenges. The Conclusion and Future perspectives sections have been well prepared and concise.
Reviewer 2 Report
I would like to thank the authors for their important topic and well written manuscript.
Comments:
1- In title the author stated “ putative adverse outcome 3 pathways.” I think this explains one of the pathways not all, as the word adverse effect applies to a remedy not a pathogenic virus, please omit or modify to “prognosis” or just “outcome”.
2- In the abstract what is “AOP framework” stands for, this is the first time it is mentioned.
3- In introduction there is no mentioning of important background elements as: the lung gut axis in COVID infection, the presence of the virus in stools (as detected by PCR) determines the outcome immunity and prognosis. These are known elements that were ignored in the introduction.
4- The authors mention “The Adverse Outcome Pathway (AOP) framework, well established in regulatory toxicology,” why are using a toxicologic paradigm in virus infection? Could you kindly clarify. If you mean this definition: “An adverse outcome pathway is a structured representation of biological events leading to adverse effects and is considered relevant to risk assessment.” Explain to the reader as just mentioning it is a toxicologic assessment is misleading in this case.
5- This sentence is not clear “Interestingly, the AOP framework provides a structured approach for the evaluation of the weight of evidence to address causality between pairs of upstream and downstream key events” please explain what do you mean by upstream and downstream? And is the word “causality” suitable for the topic?
6- The authors could shorten some of the parts in the review as it makes the reader confused when they mention other topics for as IBD and infectious colitis.
7- The figures need higher quality and more molecular details if possible.
8- The authors mentioned “If the virus is 481 hosted by bacteria of the gut microbiome, eliminating the bacterial host by appropriate antibiotics might kill the virus” , Are there any human studies on antibiotics treatment for COVID supporting this or this is a general statement? As was stated in the manuscript “Transmission electron microscopy analysis showed the presence of SARS CoV-2 particles on the surface and inside gut bacteria obtained from COVID-19 patients [112], consistent with a viral tropism for gut bacteria”
9- The effect of the COVID infection on the SCFA and balance of the Bifidobacterium vs Fermicutus is very important to be presented in a graph or table from the studies available if possible to highlight this part.
10- Could the author discuss the effect of viral shedding in stool in enhancing the immunity of the patient and the duration of the illness as mentioned in the literature and if this has any link to Gut Dysbiosis?
Author Response
Reviewer 2
Comments and Suggestions for Authors
I would like to thank the authors for their important topic and well written manuscript.
We want to thank reviewer 2 for his/her constructive comments and criticisms. We have addressed in details the comments point by point here below.
Point 1. In title the author stated “ putative adverse outcome 3 pathways.” I think this explains one of the pathways not all, as the word adverse effect applies to a remedy not a pathogenic virus, please omit or modify to “prognosis” or just “outcome”.
Response: This is a very interesting point. We understand that there may be a confusion with the term “adverse effect” of a drug as understood in Pharmacovigilance context where it relates to the toxic i.e. not desired and different than desired or target activity effects. However, the reference to Adverse Outcome Pathway (AOP) has a defined meaning in toxicology and describes a series of perturbances of the basal biological state induced by a stressor and culminating in an adverse outcome. A stressor can be a chemical for cosmetic use but also a drug or even an active ingredient in a pesticide that in addition to the target activity has a toxic or adverse activity i.e not intended for the use (https://www.oecd.org/chemicalsafety/testing/adverse-outcome-pathways-molecular-screening-and-toxicogenomics.htm).
We added capital letters in the title to Adverse Outcome Pathways to clarify that our review relies on an existing concept. We also revised the Introduction section to further describe the AOP approach as applied in toxicology, to help distinguish the use of term adverse effect from an event as understood in Pharmacovigilance (see point 4). We aim to demonstrate that the AOP approach can be applicable to biomedical and pharmaceutical sciences. Ultimately, an “adverse effect” is defined by the intended use and may be different in different regulatory contexts. In essence, the reviewer comment inspires a possibility to demonstrate that the AOP framework can also be used to better understand the underlying mechanisms of the adverse events as understood in Pharmacovigilance and help inform strategies to avoid them.
Point 2. In the abstract what is “AOP framework” stands for, this is the first time it is mentioned.
Response: The text has been changed (line 25) to spell out AOP in the abstract.
Point 3. In introduction there is no mentioning of important background elements as: the lung gut axis in COVID infection, the presence of the virus in stools (as detected by PCR) determines the outcome immunity and prognosis. These are known elements that were ignored in the introduction.
Response: We agree with the reviewer that gut-lung axis, but also gut-liver and gut-brain axes are important elements to discuss with regard to gut microbiota impact on health. We added a paragraph p.2 along with 5 additional references.
“Besides, emerging evidence demonstrated important cross-talks between the gut microbiota and many other organs via communication axes such as the gut-lung (10.1038/s41385-019-0160-6), gut-liver [20] 10.3390/microorganisms10051045) and gut-brain (10.3389/fmed.2022.813204) axes. Notably, gut dysbiosis during respiratory viral infection has been shown to worsen pulmonary symptoms (10.1016/j.jiph.2020.07.003). Similarly, gut dysbiosis and disrupted intestinal barrier can cause neurological inflammation (10.3389/fmed.2022.813204) or hepatic inflammation through translocation of endotoxins and bacteria via the portal vein (10.1002/rmv.2211).”
We agree with the reviewer that the presence of the virus in stools is also an important element to discuss. We added a paragraph in section 5.2 discussing the central role of microbiota on COVID-19.
“Despite the well-documented prevalence of GI symptoms, such as diarrhea, and the high rate of SARS-CoV-2 fecal RNA shedding, the isolation of replication-competent virus from fecal samples has not been reproducibly and systematically demonstrated (Clerbaux et al, 2022, submitted in parallel in this Special Issue). The biological, clinical, and epidemiological relevance of SARS-CoV-2 shedding remains unclear (10.1016/j.medj.2022.04.001). SARS-CoV-2 shedding in stools has been reported from one week to seven months after diagnosis (10.1016/j.medj.2022.04.001; 10.1016/S2468-1253(20)30083-2). The prolonged presence of viral RNA in feces (10.1016/j.medj.2022.04.001), but not in respiratory samples suggests that SARS-CoV-2 GI tract infection can be prolonged in a subset of individuals with COVID-19. SARS-CoV-2 infection leading to perturbation of the gut microbiome may contribute to the prolonged GI symtoms (10.1101/mcs.a006031; 10.1126/science.abm8108). Alteration of the gut microbiome persists long after a patient recovers, suggesting that the gut microbiome may play an important role in long COVID [158]. Long COVID or post-acute COVID-19 syndrome (PACS) is rapidly emerging across the globe and many studies following patients who have recovered from the respiratory effects of COVID-19 identified persistent GI sequelae, including dysbiosis (10.1016/j.eclinm.2021.101019 ; 10.1016/j.medj.2022.04.001; 10.1016/S2468-1253(20)30083-2). While the pathogenesis of long COVID is still under intense investigation, on the four current leading hypotheses (10.1126/science.abm8108), it is interesting to note that gut dysbiosis is considered as one of them (10.1097/PG9.0000000000000152; 10.1136/gutjnl-2021-325989). A comprehensive understanding of the dynamics of fecal clearance of SARS-CoV-2 RNA and its link with gut dysbiosis is currently lacking. Further studies are needed as the gut microbiota could serve as a potential prognosis indicator and could be therapeutically valuable.”
Point 4. The authors mention “The Adverse Outcome Pathway (AOP) framework, well established in regulatory toxicology,” why are using a toxicologic paradigm in virus infection? Could you kindly clarify. If you mean this definition: “An adverse outcome pathway is a structured representation of biological events leading to adverse effects and is considered relevant to risk assessment.” Explain to the reader as just mentioning it is a toxicologic assessment is misleading in this case.
Response: The second paragraph of the Introduction section was revised to describe better the universal value of the AOP approach and why it is suitable and useful to apply it to examine the mechanisms leading to gut dysbiosis in COVID-19. This study was actually realized under the CIAO project which aims to use the AOP framework to make sense of the overwhelming flux of publications related to COVID-19 pathogenesis. We added three references describing the project aiming to explore the assumption that a toxicological framework is helpful in the context of viral stressors leading to a complex disease (10.14573/altex.2102221; 10.14573/altex.2112161; 10.3389/fpubh.2021.638605).
« To contribute to decipher these mechanisms, the Joint Research Centre of the European Commission initiated an interdisciplinary project, the CIAO project, aiming to model the pathogenesis of COVID-19 using the Adverse Outcome Pathway (AOP) framework (10.3389/fpubh.2021.638605; 10.14573/altex.2102221; 10.14573/altex.2112161; www.ciao-covid.net). The AOP approach is well established in regulatory toxicology [23] but is innovatively applied here to a viral disease of high societal relevance. The project relies on the assumption that an AOP driven-organization of the knowledge will improve the integration of the tsunami of data on COVID-19 (10.3389/fpubh.2021.638605). The AOP approach does not capture all the details in a biological pathway, but aims for a pragmatic identification of successively linked key events (KE) that represent essential steps in a pathway leading to an adverse outcome [34], [35], [36], [37]. A key event describes a measurable and essential change in a biological system that can be quantified in experimental or clinical settings [33]. The AOP framework also provides a structured approach for the evaluation of the level of evidence currently available to ascertain the causal relationships between pairs of successive key events [38]. AOPs do not build on correlation between two events but gather and weight the evidence for their causal relationship. Because of this mechanistic and causal description of the pathways, AOPs help elucidate the pathophysiological mechanisms also by learning from other diseases, such as IBD or respiratory virus-related diseases presenting gut dysbiosis. Finally, an AOP integrates knowledge across the different biological levels (from molecular, cellular, tissue, organ up to organism level). While research tends to compartmentalize in silos, this pandemic calls for an interdisciplinary integration of data from the different experimental systems. Hence, the AOP approach allows structured review and organization of rapidly growing relevant in vitro, in vivo and clinical data. Assessing the evidence currently available using the AOP framework permits to identify critical inconsistencies and knowledge gaps guiding future research needs. »
Point 5. This sentence is not clear “Interestingly, the AOP framework provides a structured approach for the evaluation of the weight of evidence to address causality between pairs of upstream and downstream key events” please explain what do you mean by upstream and downstream? And is the word “causality” suitable for the topic?
Response: We revised the sentence line 83 in the revised version.
« The AOP framework also provides a structured approach for the evaluation of the level of evidence currently available to ascertain the causal relationships between pairs of successive key events [38]. AOPs do not build on correlation between two events but gather and weight the evidence for their causal relationship.»
Point 6. The authors could shorten some of the parts in the review as it makes the reader confused when they mention other topics for as IBD and infectious colitis.
Response: For chronic inflammatory diseases such as IBD and colitis, whose etiologies remain uncertain, gut dysbiosis and barrier function have been proven to have a central role. AOPs are pathways based on mechanistic understanding, they are stressor agnostic. This means that the causal relationship can be triggered by different stressors (10.1007/s00204-017-2020-z). Thus AOPs help elucidate COVID-19 pathogenesis also by learning from other diseases underlying mechanisms, such as IBD. We added a sentence in introduction line 86 in the revised version to highlight that aspect and hope to convince the reviewer of the importance of this aspect.
“Because of this mechanistic and causal description of the pathways, AOPs help elucidate the pathophysiological mechanisms also by learning from other diseases, such as IBD or respiratory virus-related diseases presenting gut dysbiosis. »
Point 7. The figures need higher quality and more molecular details if possible.
Response: We modified the figures to increase their quality and consistency. We did not add molecular details to follow the current principles ruling the graphical representation of the AOP framework (33), OECD handbook, p.19). A graphical summary of the AOP listing the KEs and their relationships is achieved using standard boxes and arrows.
Point 8. The authors mentioned “If the virus is 481 hosted by bacteria of the gut microbiome, eliminating the bacterial host by appropriate antibiotics might kill the virus” , Are there any human studies on antibiotics treatment for COVID supporting this or this is a general statement? As was stated in the manuscript “Transmission electron microscopy analysis showed the presence of SARS CoV-2 particles on the surface and inside gut bacteria obtained from COVID-19 patients [112], consistent with a viral tropism for gut bacteria”
Response: We thank the reviewer for the valuable comment. The section 4.3 Potential implication for disease management p.11 of the revised version has been modified accordingly.
“This proposed mechanism might have a direct effect on human health. If the virus is hosted by bacteria of the gut microbiome, eliminating the bacterial host by appropriate antibiotics might kill the virus [132]. The efficacy of some antibiotics (like rifaximin and azithromycin) in reducing viral RNA load to negligible levels in in vitro fecal microbiota cultures obtained from stool samples of SARS-CoV-2 positive individuals has been reported [13] (10.12688/f1000research.54306.2). However, a better understanding via the AOP concept of these complex interactions makes prevention or adequate therapeutic interventions mechanism-based taking into account different modulating factors [124]. In this regard, multidisciplinary approaches that couple tests on the efficacy of antimicrobials with proteomic and electron microscopy image analyses would be beneficial to shed light on the potential viral tropism for gut bacteria of SARS-CoV-2.”
Point 9. The effect of the COVID infection on the SCFA and balance of the Bifidobacterium vs Fermicutus is very important to be presented in a graph or table from the studies available if possible to highlight this part.
Response: The ratio between the Bifidobacterium and Fermicutus as well as the level of SCFA (producing bacteria) are widely accepted as having important influence in maintaining normal intestinal homeostasis. Alterations of such ratio and level is indicative of dysbiosis. Presenting the effect of SARS-CoV-2 infection on those indicators within a table helps to support that gut dysbiosis occurs in SARS-CoV-2 infected patients (correlation between initial step and late adverse outcome) but does not inform about the underlying mechanisms. This review mentions the main articles supporting evidence for gut dysbiosis as an adverse outcome. We aim to present these data in a table in the AOP Wiki (https://aopwiki.org/events/1954) as this online and open access platform can be constantly updated when new data become available
Point 10. Could the author discuss the effect of viral shedding in stool in enhancing the immunity of the patient and the duration of the illness as mentioned in the literature and if this has any link to Gut Dysbiosis?
Response: The extended presence of viral RNA in feces, but not in respiratory samples, and its association with GI symptoms suggests that SARS-CoV-2 potential infection of the GI tract can be prolonged in a subset of individuals with COVID-19. However, a comprehensive understanding of the dynamics of fecal clearance of SARS-CoV-2 RNA and its link with gut dysbiosis is of crucial importance but lacking. Studying gut microbiota perturbations in COVID-19 from a mechanistic point of view could enhance our understanding of COVID-19 pathophysiology. To address the reviewer comment, we added a whole paragraph in section 5.2 (as mentioned in point 3) and a sentence in section 5.3.
Section 5.2 Central role of gut in COVID-19
“Despite the well-documented prevalence of GI symptoms, such as diarrhea, and the high rate of SARS-CoV-2 fecal RNA shedding, the isolation of replication-competent virus from fecal samples has not been reproducibly and systematically demonstrated (Clerbaux et al,). The biological, clinical, and epidemiological relevance of SARS-CoV-2 shedding remains unclear (10.1016/j.medj.2022.04.001). SARS-CoV-2 shedding in stools has been reported from one week to seven months after diagnosis (10.1016/j.medj.2022.04.001; 10.1016/S2468-1253(20)30083-2). The prolonged presence of viral RNA in feces (10.1016/j.medj.2022.04.001), but not in respiratory samples suggests that SARS-CoV-2 GI tract infection can be prolonged in a subset of individuals with COVID-19. SARS-CoV-2 infection leading to perturbation of the gut microbiome may contribute to the prolonged GI symtoms (10.1101/mcs.a006031; 10.1126/science.abm8108). Alteration of the gut microbiome persists long after a patient recovers, suggesting that the gut microbiome may play an important role in long COVID [141]. Long-term sequelae of COVID-19, post-acute COVID-19 syndrome (PACS) or long COVID, are rapidly emerging across the globe and many studies following patients who have recovered from the respiratory effects of COVID-19 identified persistent GI sequelae, including dysbiosis (10.1016/j.eclinm.2021.101019 ; 10.1016/j.medj.2022.04.001; 10.1016/S2468-1253(20)30083-2). While the pathogenesis of long COVID is still under intense investigation, on the four current leading hypotheses (10.1126/science.abm8108), it is interestingly to note that gut dysbiosis is considered as one of them (10.1097/PG9.0000000000000152; 10.1136/gutjnl-2021-325989). A comprehensive understanding of the dynamics of fecal clearance of SARS-CoV-2 RNA and its link with gut dysbiosis is currently lacking. Further studies are needed as aside from serving as a potential prognosis indicator, the gut microbiota could be therapeutically valuable.”
Section 5.3. FMT treatment
“So far, FMT has been remarkably successful in the treatment of Clostridium difficile infection but much less in treating other conditions, such as IBD or metabolic disorders. COVID-19 being an infectious disease and not an inflammatory disorder, it could be more successful [162]. However, COVID-19 could potentially be transmitted via FMT particularly from asymptomatic donors who tested negative for the presence of the virus in their respiratory tract but positive in their stools (10.3390/ijms22063004). No cases of COVID-19 transmission through FMT have been reported so far, but only FMT products generated from stools donated before December/November 2019 were used according to the FDA and Hong Kong recommendations, respectively.”
Round 2
Reviewer 1 Report
The authors have cautiously replied and fully revised the manuscript as the suggestions and comments of the reviewer. Therefore, this work should be recommended for publication in this journal.